# Light-FER: A Lightweight Facial Emotion Recognition System on Edge Devices

**DOI:** 10.3390/s22239524

**Published:** 2022-12-06

**Authors:** Alexander M. Pascual, Erick C. Valverde, Jeong-in Kim, Jin-Woo Jeong, Yuchul Jung, Sang-Ho Kim, Wansu Lim

**Affiliations:** 1Department of Aeronautics, Mechanical and Electronic Convergence Engineering, Kumoh National Institute of Technology, Gumi 39177, Republic of Korea; 2Department of Data Science, Seoul National University of Science and Technology, Seoul 01811, Republic of Korea; 3Department of Computer Engineering, Kumoh National Institute of Technology, Gumi 39177, Republic of Korea; 4Department of Industrial Engineering, Kumoh National Institute of Technology, Gumi 39177, Republic of Korea

**Keywords:** edge device, facial emotion recognition, model compression, Xception

## Abstract

Facial emotion recognition (FER) systems are imperative in recent advanced artificial intelligence (AI) applications to realize better human–computer interactions. Most deep learning-based FER systems have issues with low accuracy and high resource requirements, especially when deployed on edge devices with limited computing resources and memory. To tackle these problems, a lightweight FER system, called Light-FER, is proposed in this paper, which is obtained from the Xception model through model compression. First, pruning is performed during the network training to remove the less important connections within the architecture of Xception. Second, the model is quantized to half-precision format, which could significantly reduce its memory consumption. Third, different deep learning compilers performing several advanced optimization techniques are benchmarked to further accelerate the inference speed of the FER system. Lastly, to experimentally demonstrate the objectives of the proposed system on edge devices, Light-FER is deployed on NVIDIA Jetson Nano.

## 1. Introduction

Facial emotion recognition (FER) is one of the most popular subjects of study in the field of emotion recognition. The FER system is an emotion recognition system that analyzes human facial expressions to determine which emotion is being expressed. The following are some of the most prevalent human emotions used in FER: (1) anger, (2) disgust, (3) fear, (4) happiness, (5) sadness, (6) surprise and (7) neutral. The sophisticated implementation of FER, which normally provides a wide range of accuracy, is due to these seven emotions. For example, the highest accuracy in [1] is found in happiness, while the lowest accuracy is found in fear, resulting in a low total accuracy of 76%. In addition, [2] used K-nearest neighbor (KNN) and artificial neural network (ANN) to investigate alternative supervised classification task methods for FER. However, due to the related low accuracy from distinct emotion categories, both KNN-based and ANN-based FER approaches produce low accuracy results (54.16% and 66.66%, respectively). To address the issue of accuracy, deep learning (DL)-based methods have been proposed to enhance the overall accuracy of FER algorithms by up to 90% [3]. As a result, researchers are concentrating their efforts on constructing deeper networks to detect more complex features and improve the accuracy of the FER system, trying to find significant face patterns that can improve the accuracy of the FER system to more than 95% [4]. On the other hand, other studies [5,6] developed FER systems that can handle both static/still images (laboratory-controlled environment) and in-the-wild conditions which are more natural and spontaneous than those of a laboratory-controlled environment. In another study, ref. [5] proposed a novel deep convolutional neural network framework to extract holistic features in identifying facial expressions while adopting a gravitational force-based edge descriptor to fetch low-level local features. Furthermore, ref. [6] proposed a simple CNN network to classify static expressions that perform well even on a small dataset which is specifically designed to learn detailed local features (e.g., eyes and mouth corners) that are exhibited by different facial expressions in face images. In [7], a texture-based feature-level ensemble parallel network was proposed for FER that uses multi-scale convolutional and multi-scale residual block-based DCNN, which prevents the deep network from having insufficient training data and improves the recognition rate while lowering computational complexity. Both proposed methods can accurately and precisely recognize facial expressions, which outperforms up to 25 baseline methods in terms of feature extraction analysis, classification accuracy and precision. However, in exchange for improved accuracy outcomes, the inference speed and computing costs of FER system are frequently overlooked.

To deal with the challenges of deep learning models, some researchers have proposed the creation of lightweight FER models with lower resource requirements and faster inference speed. Several studies use state-of-the-art pre-trained DL models, such as VGG-Face and VGG16 and Deep Face Convolutional Neural Network (CNN), to create transfer learning-based lightweight FER models. Their studies, however, only compare which pre-trained models can provide both high accuracy and a simpler architecture when compared to other existing models. That is, the original structure of the pre-trained models is not reduced and thus there is no relative improvement in the results.

In this paper, we propose Light-FER, a lightweight FER system developed by optimizing the hardware and software components of the Xception model using various compression methods. The contributions of this paper are summarized as follows:To reduce computational costs, pruning is used to remove unimportant and redundant connections within the architecture of the FER model.To reduce memory usage and increase the inference speed of the FER system, quantization is used to save the lightweight FER model in a lower precision format.To increase inference speed while maintaining low computational costs, a DL compiler will be used to redesign the FER model in order to take advantage of the available hardware of the device.To verify the effectiveness of the proposed FER system on edge devices, Light-FER is deployed on NVIDIA Jetson Nano.

The remainder of this paper discusses in detail the development of the FER system, various network compression methods, the proposed methodology and the conclusion.

## 2. Related Works

In this section, the development of a facial emotion recognition (FER) system is thoroughly discussed. This is followed by a discussion of various network compression methods that can be used to improve the inference performance of the FER system.

### 2.1. Development of FER System

The FER systems are divided into two tasks: (1) face detection and (2) facial emotion classification. The human face is detected and fed into an algorithm in the FER system, which analyzes patterns and classifies facial expressions. The emergence of cutting-edge face detection algorithms resulted in the development of these FER systems.

In the face detection task, multi-task cascaded convolutional neural network (MTCNN) [8] and Deep alignment network (DAN) [9] have been recently used for face alignment and detection to ensure high accuracy. As shown in Figure 1, DAN generates a total of 68 facial landmarks, whereas MTCNN only generates five. However, because both MTCNN and DAN are built with multiple stages of CNNs, they require high computational costs, resulting in slow inference speed during real-time implementation. In this paper, a pre-trained 68-facial landmark detector trained by Dlib [10] in the iBUG 300-W dataset is used. It produces similar results to DAN but at a lower computational cost.

In the facial emotion classification task, several facial expression or FER models have been developed to achieve higher accuracy results. CNN architectures are typically used to leverage deeper networks. The number of parameters in a CNN architecture is increased by using more layers and smaller kernels, which can learn more complex patterns. However, DL-based FER models suffer from a low inference speed issue due to the high computational cost requirement. To address this issue, several researchers worked on developing lightweight FER models to allow low computational cost and fast inference implementation. The key to lightweight FER models is significant reduction in the parameters.

### 2.2. Network Compression Methods

DL-based FER systems suffer from high resource requirements when applied to edge devices. To address these challenges, several network compression methods have been studied and performed. These methods, namely, pruning, quantization and DL-based compiler optimizations, can achieve faster inference speed, reducing computational costs and complexity of CNN models. The goal is to simplify the model without considerably compromising its accuracy.

Pruning refers to the effective search for and elimination of unimportant and redundant connections within the CNN model [11]. Weights, filters and channels are the connections in the CNN model that contribute to its enormous complexity. Pruning can permanently remove a number of less significant connections within a CNN model in a fine-grained nature. It can take advantage of the hardware parallelism of the device at the expense of significantly reducing the storage footprint of the model while securing zero or negligible accuracy loss. As a result, the memory and computational costs of the model will be reduced during implementation. However, there are challenges in finding the compression hyperparameter in each layer of the CNN model that causes a general problem in applying pruning to a model. To effectively optimize the hyperparameters, several approaches can be used. One of these approaches is known as a heuristic method or manual tuning of the hyperparameter per layer of the CNN architecture. Another method is black box optimization, also known as Bayesian optimization, which is an automatic hyperparameter-search approach. These approaches, however, are inefficient because they rely solely on repeated trial-and-error procedures to determine the optimal hyperparameters in a layer-wise manner, which frequently results in a significant accuracy loss. Meanwhile, quantization refers to the conversion of the bit representation of each weight of the CNN architecture into a lower precision format [12]. Because CNN models are initially stored in 32-bit floating-point (FP32) format, they can be quantized to FP16 or even lower integer (INT) formats such as INT8 and INT4. As a result, memory consumption is reduced while inference speed is increased. The general pruning and quantization process is demonstrated in Figure 2. Finally, a DL compiler can be used to leverage with the hardware architecture of the model [13]. It can optimize models by redesigning and compiling its architecture, so that hardware optimizations for faster inference are easily accessible. It includes advanced performance optimization, variable computation graph optimization, tensor optimization and support for half precision formats. These DL compilers have similar processes for optimizing CNN models. In this paper, a DL compiler is utilized to compile the Xception model into an efficient end-to-end framework in order to improve the inference performance of the FER system.

## 3. Proposed Lightweight FER System

The proposed lightweight FER system is illustrated in Figure 3. As shown in the figure, the Light-FER system includes three main processes: (1) face detection in the input image, (2) network compression of the FER model and (3) facial emotion classification.

First, we used a pre-trained landmark detector from the Dlib library to identify 68 key points or facial landmarks marked at specific x and y coordinates in the human face.

The key points define the area around the human face, including the brows, eyes, nose, mouth, chin and jaw. The Dlib 68-landmark face detector also provides the coordinates of a bounding box enclosing the human face. It is trained on the iBUG 300-W dataset, which was created by the Intelligent Behavior Understanding Group (iBUG) at Imperial College London. The iBUG 300-W dataset contains 68 facial landmarks and bounding box annotations from “in-the-wild” images collected from the internet. It contains over 4000 static images, each with a single face in a variety of poses, expressions and illuminations. The trained model is tested on a 300-W test dataset that includes “in-the-wild” images, 300 indoors and 300 outdoors, with multiple faces and large facial variations in each image. Based on a 300-W test dataset, the pretrained face detection model has an efficient inference performance in detecting faces. As a result, in this paper, the Dlib 68-landmark face detector is used to extract useful patterns from various types of emotions in human faces. For an accurate FER system, it can provide an efficient calculation to distinguish one specific emotion from another. Second, in order to apply network compression methods to the FER model, we used pruning, quantization and a DL compiler with the Xception model to create a lightweight FER system. As trained and tested on the FER2013 dataset, an Xception model outperforms existing CNN models (e.g., VGG-Net and ResNet-50) in terms of accuracy and parameters. However, it still has issues with high computational costs, high memory usage and slow inference speed during FER implementation. To address these issues, we used several compression methods that targeted both software and hardware components of the FER system.

Initially, we used the pruning method to reduce the total number of non-zero Xception model parameters by removing redundant and insignificant parameters. This compression method will significantly reduce the computational costs of the model. The compression ratio of the pruned CNN architecture relative to the uncompressed is typically given by
(1)$$R=\frac{\sum_{i=1}^{l}U_{w,f,c}^{\left(i\right)}}{\sum_{i=1}^{l}P_{w,f,c}^{\left(i\right)}}$$
where R is the compression ratio between uncompressed (U) and pruned models (P); $U_{w,\;f,\;c}$ (i) and $P_{w,\;f,\;c}$ (i) represent the total number of weights (w), filters (f) and/or channels (c) up to layer L of the uncompressed and pruned CNN architecture, respectively.

Then, we used a quantization method to reduce memory usage and increase inference format from speed of the Xception model by converting its high precision FP32 to FP16 or INT8 low precision format. The quantization of general weights is computed by
(2)$$W_{b}=\partial \left(S_{b}W_{fp}\right)\epsilon \left[-\left(\frac{2^{b}-1}{2}\right),\left(\frac{2^{b}-1}{2}\right)\right]$$
(3)$$S_{b}=\frac{2^{b}-1}{2M_{w}}$$
(4)$$M_{w}=max\left(abs\left(W_{fp}\right)\right)$$
where $W_{b}$ represents the quantized weight tensor, b is the desired lower precision format, $S_{b}$ is the quantization scale factor, $W_{fp}$ is the original weight tensor in higher-precision floating-point format and $M_{W}$ is the absolute maximum weight from $W_{fp}$. The quantized weight tensor is obtained by the product of the quantization scale factor and each weight from the original weight tensor rounded to the nearest integer. The quantized weights are bounded by the symmetrical dynamic range of the desired lower-precision format. This will reduce the memory usage of Xception model by more than 50% and slightly increase its inference speed.

Pruning and/or quantization are generally difficult to implement since it is difficult to find compression hyperparameters in each layer of the CNN model. Several existing approaches (e.g., heuristic method, Bayesian optimization) are ineffective in determining which hyperparameters are used to compress the network that typically leads to a large accuracy loss. To address this issue, we used a constrained approach to pruning and quantization methods in order to reduce any potential accuracy degradation after optimization. To eliminate the risk of low accuracy results, this approach will not consider any compression hyperparameters.

Finally, we used a DL compiler to further optimize and accelerate the inference speed of the Xception model. Each framework (e.g., PyTorch, Keras and TensorFlow) has its own distinct representation of a computation graph, which frequently leads to limitations when moving a model from one framework to another. We solved this problem by implementing an Open Neural Network Exchange Format (ONNX), which is intended to standardize network layer definitions and support the majority of deep learning model formats. In this paper, a Keras framework is used to train and optimize the Xception model. The network graph is then rebuilt using an existing Keras to ONNX converter with the equivalent operators in ONNX format. This will enable the ONNX Runtime, a performance-focused inference optimizer, to automatically use the host device’s hardware accelerators and runtime. As a result, the performance of the model improves. This inference engine divides the execution graph into subgraphs and runs each subgraph on the most efficient execution provider available, such as CUDA or TensorRT. By applying pruning, quantization and DL compiler, a lightweight FER model is developed, as shown in Figure 3.

## 4. Results and Discussion

In this section, we perform several experiments to evaluate the effectiveness of our proposed Light-FER. First, the implementations, including experimental platforms, will be introduced in detail. Next, we verify the actual test results of our proposed method. Then, we compare the Light-FER to other existing FER models in terms of training and test accuracy using FER2013 dataset. Then, we validate the viability of the proposed method based on its hardware performance as compared to other FER models. Finally, to observe the best possible output of the proposed Light-FER, we perform a benchmarking experiment using different DL compilers as deployed on NVIDIA Jetson Nano.

The experiments are originally implemented on Keras framework. We train and compress the models on the computer with Intel Xeon CPUs of 2.2 GHz and an NVIDIA Tesla V100 GPU. Then, the models are initially deployed on a local PC with Intel Core i5-10300H CPU at 2.50 GHz and 16 GB memory. After that, to test the performance of the Light-FER on edge devices, we deployed the proposed method on NVIDIA Jetson Nano. The NVIDIA Jetson Nano is a low-power edge device that consumes less than 10 W. The operating system of this edge device is Ubuntu 18.04, with CUDA 10.0 and cuDNN 7.6.

In order to verify the effectiveness of the Light-FER, we perform an actual test on a local PC and observe if the result is correct for each emotion category. Table 1 shows the accuracy results when the Light-FER model is used. As can be seen, all results are correct for each emotion showing different accuracy number. The accuracy results are relatively high for anger, disgust, happy and surprise, reaching more than 95%, followed by fear at 84%. The sad and neutral classes have low accuracies at about 40% and 50%, respectively. Nonetheless, these results indicate that the Light-FER can be used to accurately recognize human emotion based on facial patterns.

Table 2 compares the accuracy of the proposed Light-FER to other existing FER models. The proposed model has the highest test accuracy result, as shown in the table, reaching 69%. Other models, such as VGG-Net, ResNet-50 and CNN, suffer from overfitting because their accuracy on the train set is close to 100%, but their accuracy on the test set is only around 60%. This implies that the proposed model outperforms other FER models in an unconstrained environment and still provides decent accuracy despite having network modifications in the network.

Table 3 displays the hardware performance of the proposed lightweight FER model. As shown in the table, the resulting parameters of the proposed method are significantly lower than the other models, such as VGG-Net, ResNet-50 and simple CNN, because of the applied pruning method that eliminates unimportant weights. This is in addition to the advantage of using Xception model that uses Global Average Pooling rather than a fully connected layer. Furthermore, the computational cost of the proposed model in terms of CPU usage is only second to CNN; however, the significant difference in memory usage makes the proposed model (3.1%) superior to CNN (12%).

Lastly, the overall performance of the proposed method is evaluated on NVIDIA Jetson Nano by measuring the inference speed and hardware parameters on different frameworks at full and half precision format. The comparison of the evaluation metrics is given in Table 4. We select Keras at FP32 as the baseline in this comparative analysis. It can be seen that, among the used frameworks, TensorRT has the best overall performance as deployed on NVIDIA Jetson Nano. The fastest inference speed is measured at TensorRT FP32 of about 5.5 FPS. The power consumption, on the other hand, is relatively similar to all experiments at around 3300 mW. The lowest memory usage, 1.06 GB, is measured in TFLite framework, but its inference speed is 12% slower than of TensorRT.

The above experiments show that our proposed method sufficiently achieved the objective of developing an FER system with lower computational costs and memory usage and relatively faster inference speed as deployed on edge devices having limited resources.

## 5. Conclusions

We proposed a lightweight FER system in this paper that achieves efficient inference performance during real-time implementation. We demonstrated experimentally that using network compression methods such as pruning and quantization can significantly reduce the computational costs and memory usage of the Xception model without affecting its accuracy results. Using a DL compiler, such as TensorRT, can further improve the inference speed of the model. The experimental results show that the proposed lightweight FER system, particularly in terms of real-time inference, which employs the Dlib 68-landmark face detector, outperforms other existing FER models. In particular, when compared to other existing FER systems, the proposed method has lower CPU and memory usage (17.50% and 3.1% in local PC; 33.6% and 57.5% in NVIDIA Jetson Nano) and a relatively high inference speed of 27 fps in local PC and 5.51 fps in NVIDIA Jetson Nano. These experimental results prove that the proposed lightweight FER system can be efficiently implemented on edge devices with low computational and memory capability. In future work, other edge devices, such as NVIDIA Jetson TX2 and Xavier, can be used to further improve the performance of the proposed lightweight FER system.

Further study is necessary, as the face detector (dlib 68-facial landmark detector) used in this paper has its limitations in detecting faces that are at a steep angle. The FER2013 dataset is collected in in-the-wild condition which includes some faces with a pose angle that is greater than 45 degrees. In order to overcome this situation, a face detector method should be developed and configured to extract a total of 68 facial landmarks that can handle facial poses at a steep angle while not consuming high power resources, or can be efficiently deployed on edge-type devices. Moreover, our proposed lightweight FER system was only trained and tested using one dataset (FER2013), so future works will include applying it to more than one dataset to allow generalizability of our method.

## Figures and Tables

**Figure 1 sensors-22-09524-f001:**
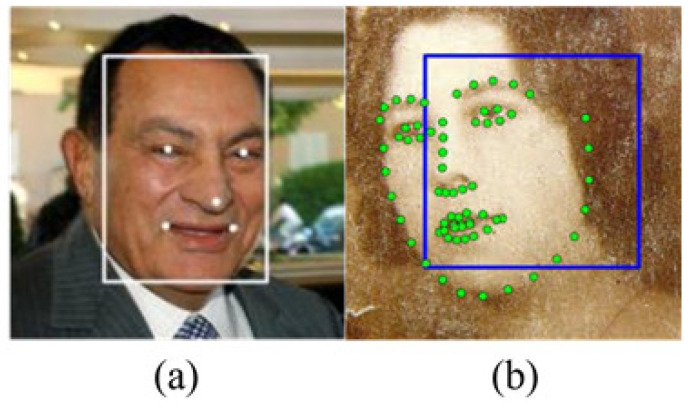
Comparison of (**a**) MTCNN and (**b**) DAN.

**Figure 2 sensors-22-09524-f002:**
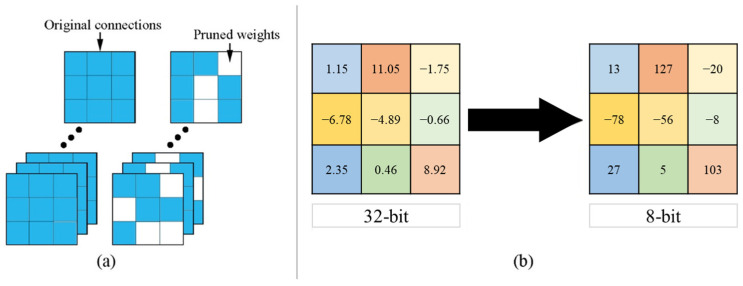
General process of (**a**) pruning and (**b**) quantization. Note that pruning shows an example of removing weights and quantization shows an example of converting 32-bit floating-point number to 8-bit integer.

**Figure 3 sensors-22-09524-f003:**
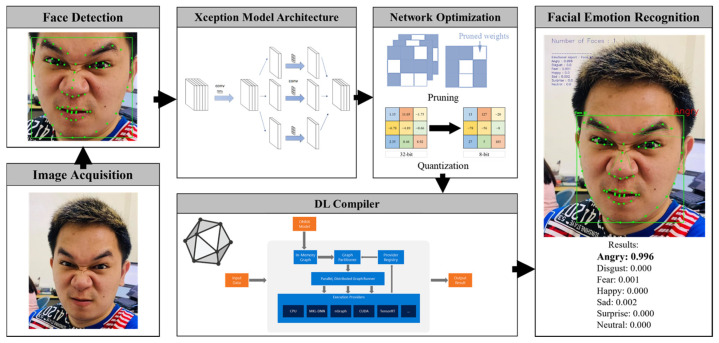
Schematic diagram of the proposed Light-FER system.

**Table 1 sensors-22-09524-t001:** Actual test results of the proposed Light-FER.

Light FER	Angry	Disgust	Fear	Happy	Sad	Surprise	Neutral
Angry	**0.996**	0.000	0.001	0.000	0.002	0.000	0.000
Disgust	0.002	**0.997**	0.000	0.000	0.000	0.000	0.000
Fear	0.003	0.000	**0.839**	0.000	0.128	0.027	0.003
Happy	0.001	0.000	0.001	**0.992**	0.000	0.005	0.002
Sad	0.160	0.001	0.029	0.003	**0.414**	0.001	0.391
Surprise	0.004	0.000	0.035	0.003	0.000	**0.957**	0.000
Neutral	0.156	0.000	0.050	0.008	0.271	0.004	**0.511**

**Table 2 sensors-22-09524-t002:** Comparison of different FER models based on training and test accuracy results.

Model	Train (%)	Test (%)
VGG-Net [14]	98.98	59.32
ResNet-50 [14]	98.87	57.48
Simple CNN [14]	99.70	58.90
HOG + CNN [15]	-	61.86
Improve Inception [16]	-	66.41
Network from [17]	-	66.40
CNN + Softmax [18]	-	65.03
ShallowNer [19]	-	63.49
**Light-FER**	**82.35**	**69.87**

**Table 3 sensors-22-09524-t003:** Hardware Performance of Different FER Models.

**Model**	**CPU (%)**	**MEM (%)**	**FPS**
VGG-Net [14]	22.3	1.6	11
ResNet-50 [14]	52.1	1.3	15
Simple CNN [14]	11.5	6.1	19
**Light-FER**	**18.4**	**1.5**	**27**

**Table 4 sensors-22-09524-t004:** Overall performance of the proposed FER system using different deep learning frameworks on Jetson Nano.

Framework	Format	FPS	P (mW)	CPU (GB)	GPU (GB)	Total (GB)
Keras	FP32	2.5	3215	2.1	1.3	3.4
TFLite	FP32	4.85	3310	0.844	0.216	1.06
TFLite	FP16	4.87	3300	0.863	0.216	1.08
ONNX	FP32	5.21	3480	2.6	1.3	3.9
ONNX	FP16	4.73	3470	2.9	1	3.9
TensorRT	FP32	5.43	3390	1.9	0.467	2.37
TensorRT	FP16	5.51	3360	1.9	0.467	2.37

## Data Availability

Not applicable.

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
