# Peer review of "Light-FER: A Lightweight Facial Emotion Recognition System on Edge Devices"

_sensors, 2022, doi:10.3390/s22239524_

Round 1

Reviewer 1 Report

The authors have proposed a FER system that has lower computational parameters. They have tried to write it well, but the novelty of the work is not clear. It looks like they have just used some developed methods and applied them to a dataset. The following are the comments that need to be addressed before publication:

1.      In the Introduction, it would be great if the authors could mention the FER systems developed for images/videos captured in the real world and those for laboratory data. 

2.      In the “Related work” section, the authors normally just explain the methods that existed in the literature and discuss the advantages and limitations of these methods. In this paper, the authors express their utilized techniques a lot in this section. For example, in section 2.2, the first paragraph explains the used compression method while it should be explained in the “Methodology” section or briefly pointed out at the end of this Section.

3.       It seems that the authors need to restructure their paper. In Section 3, they discuss the existing methods for pruning while they could be explained in Section 2. Two sections “Related work” and “Proposed light-weight FER system” are mixed, they need to be separated.

4.      The fear accuracy is around 84%, but page 7 states it is more than 95%.

5.      Trying one dataset is not enough to conclude a method outperforms others. The experiments could be extended to include at least one more dataset.

6.      Dlib facial detector is not good enough in detecting faces that are at a steep angle. How have the authors managed this situation? Because the FER2013 dataset includes some faces with a pose angle is > 45 degree. Some authors [1] have tried to use other methods to handle this problem.

[1] N. Samadiani, G. Huang, Y. Hu and X. Li, "Happy Emotion Recognition From Unconstrained Videos Using 3D Hybrid Deep Features," in IEEE Access, vol. 9, pp. 35524-35538, 2021, doi: 10.1109/ACCESS.2021.3061744.

Reviewer 2 Report

The paper proposed a lightweight facial emotion recognition system, called Light-FER. This model achieves the purpose of improving the operation efficiency by modifying the Xception model. However, the innovation of the paper is not high, and the overall description of the model is not clear enough, such as DL compiler, ONNX, etc.

Reviewer 3 Report

My comments are the following:

1. I agree with the authors that FER is imperative for the recent advanced artificial intelligence (AI) applications to realize better human-computer interactions. However, it is not a new problem. Many advancements have already been made. Most of the existing works perform well in controlled environments. However, the performance of these systems is not satisfactory in the wild. So, the authors should stress the wild conditions.

2. Small improvements are made to an existing architecture of Xception. So, the contribution is limited.

3. The authors have considered a few models for comparison, which is not sufficient in today's context. For comparison purposes, the authors consider more advancement methods from the domain.

4. The authors should discuss the following papers in the proper places:

a) Facial expression recognition using local gravitational force descriptor-based deep convolution neural networks

b) FER-net: facial expression recognition using deep neural net

c) LieNet: A Deep Convolution Neural Networks Framework for Detecting Deception

5. There are many benchmark databases that are freely available to show the effectiveness of the proposed method. So, the authors should consider some of the available databases as one database is not enough in today's context.

6. Digitizer result of Abstract.

Try to put some digitizer results of the most important in this paper in the Abstract. The abstract is the first image of readers can catch their eye or not.

7. The authors should showcase some failure cases with proper justifications, and why the proposed method is not enough. How these limitations can be overcome should mention in the conclusion section.

Round 2

Reviewer 1 Report

The revised paper sounds better and the contributions are clearer in this version. There is only one concern regarding the generalizability of the method and the authors could show its performance by applying at least one more dataset.

Reviewer 2 Report

To be honest, I am not very convinced by the conclusions of the paper. According to the paper, the author has done a lot of model slimming operations, including model pruning, ONNX, etc. But there is no mention of what has been done to improve performance. After doing so much pruning, this model has the best performance on the test set compared to several other models (although the compared models are not the latest and greatest), which I am not very sure about .

Reviewer 3 Report

Discuss the paper "FLEPNet: Feature Level Ensemble Parallel Network for Facial Expression Recognition" in the proper place.
